# The Tomographic Study and the Phenotype of Wormian Bones

**DOI:** 10.3390/diagnostics13050874

**Published:** 2023-02-24

**Authors:** Ali Al Kaissi, Sergey Ryabykh, Farid Ben Chehida, Hamza Al Kaissi, Susanne Gerit Kircher, Martin J. Stransky, Franz Grill

**Affiliations:** 1Pediatric Department, Orthopedic Hospital of Speising, 1130 Vienna, Austria; 2National Medical Research Center for Traumatology and Orthopedics n.a. G.A. Ilizarov, 640014 Kurgan, Russia; 3Ibn Zohr Diagnostic Institute of Radiology, Tunis 1003, Tunisia; 4Surgical Outpatient Clinic of Landesklinikum Hospital, 3400 Klosterneuburg, Austria; 5Center of Pathobiochemistry and Genetics, Medical University of Vienna, 1090 Wien, Austria; 6Department of Neurology, Yale School of Medicine (USA), New Haven, CT 06510, USA; 7Policlinic at Národní, 110 00 Prague, Czech Republic

**Keywords:** wormian bones, radiology, tomography, phenotype, overly stretched pastry, bulging of the occipital lobe of the cerebrum, basilar invagination

## Abstract

Background: We describe patients who were recognized via conventional radiographs of the skull as manifesting wormian bones. Wormian bones are not a specific diagnostic entity and can be seen in variable forms of syndromic disorders. Materials and Methods: Seven children and three adults (of 10–28 years) were seen and diagnosed in our departments. The principal complaints for the pediatric and adult group were: ligamentous hyperlaxity, a history of delayed walking and occasional fractures, which later in life started to manifest a constellation of neurological symptoms such as nystagmus, persistent headache, and apnea. Conventional radiographs were the first traditional tools used to recognize wormian bones. We performed 3D reconstruction CT scans to further understand the precise etiology and the nature of these wormian bones and attempted to connect them with a broad spectrum of unpleasant clinical presentations. Our group of patients was consistent with the phenotypic and genotypic diagnoses of osteogenesis imperfecta type I and type IV as well as patients with multicentric *carpotarsal osteolysis* syndrome. Results: Three-dimensional reconstruction CT scan of the skulls confirmed that these worm-like phenotypes are in fact stemmed from the progressive softening of the sutures. The overall phenotype of the melted sutures is akin to overly stretched pastry. The most concerning sutures in this pathological process are the lambdoid. The overstretching of the lambdoid sutures was responsible for the development of sub-clinical basilar impression/invagination. Patients with certain forms of skeletal dysplasia such as osteogenesis imperfecta type I and *IV manifested the heterozygous mutation of COL1A1/COLA2, shown as typical overstretching of the sutures*. Similarly, patients with *multicentric carpotarsal osteolysis* syndrome with a heterozygous missense *mutation of MAFB also manifested the phenotype of overly stretched pastry along the skull sutures*. Conclusion: What we encountered via 3D reconstruction CT scan in our group of patients was entirely different than the traditional description that can be found in all relevant literature of the last decades. The worm-like phenomenon is in fact a pathological sequel occurring as a result of a progressive softening of the sutures, which results in the overstretching of the lambdoid sutures, a pathological process roughly similar to an overly stretched soft pastry. This softening is totally connected to the weight of the cerebrum (the occipital lobe of the cerebrum). The lambdoid sutures represent the weight-bearing zone of the skull. When these joints are loose and soft, they adversely alter the anatomical structures of the skull and lead to a highly hazardous derangement of the craniocervical junction. The latter causes the pathological upward invasion of the dens into the brain stem, leading to the development of morbid/mortal basilar impression/invagination.

## 1. Introduction

The traditional definition of wormian bones is small bones that are often found within the sutures and fontanelles of the skull. Some instances are often considered to be a simple anatomical variant. Previous studies concluded that around 8%-15% of the population has at least one wormian bone [1,2]. In patients with significant pathologies, there are least ten wormian bones larger than around 6 mm × 4 mm arranged in a mosaic-type pattern. The skull itself is composed of several flat bones that fuse together after birth. These sites of fusion are the bony sutures in which wormian bones most commonly occur [2,3]. 

Wormian bones are more commonly seen in patients with several types of bone dysplasia. Osteogenesis imperfecta is the most common type of bone dysplasia, in which wormian bones represent one of the main diagnostic features. Wormian bones can be also encountered in a long list of heritable syndromes [1,2,3]. Weissbach et al. in two occasions described fetal wormian bones and their clinical outcomes in prenatally diagnosed fetuses [4,5].

Osteogenesis imperfecta (OI) is a clinically and genetically heterogeneous group of heritable disorders of the connective tissue characterized by reduced bone mass (osteopenia) with associated bone fragility. The resulting skeletal manifestations are due to a generalized deficiency in the development of both membranous and endochondral bone and include markedly thin calvarium with delayed closure of the fontanelles and the sutures and excessive wormian bone formation. OI type I is a purely qualitative collagen defect, whereas OI of type II, III, and IV are qualitative and quantitative alterations in collagen synthesis [6,7].

Type I osteogenesis imperfecta (OMIM 166200) is the commonest form of osteogenesis imperfecta and is inherited as an autosomal dominant condition. Affected individuals may have blue sclerae with a tendency to fractures of the long bones, although healing occurs without deformity. Cells from individuals with type I (OI) secrete about half the normal amount of type I procollagen. More than 95% of individuals with OI are heterozygous for mutations in either of the two type I collagen genes, *COL1A1* and *COL1A2*. The majority of mutations associated with OI occur in the type I collagen encoding genes and give rise to an autosomal dominant form of the disease. Novel mutations in seven other genes involved in collagen assembly and processing and in two genes involved in cellular differentiation have recently been associated with autosomal recessive forms of OI [6].

Osteogenesis imperfecta type IV (OMIM 166220) is similar to type I, but the sclerae might be normal and mild short stature with limb bowing is a feature. There have been extensive studies regarding the true nature of wormian bones and why they occur. Silence and Glorieux described wormian bones in association with osteogenesis imperfecta type I and IV. Platybasia and basilar impression/basilar invagination may occur because of bone softening in patients with osteogenesis imperfecta [7,8,9]. Semler et al. discussed wormian bones in osteogenesis imperfecta in correlation with phenotypic and genotypic characteristics [10].

Sykes et al. found linkage to the COL1A2 locus in eight pedigrees [11]. Lund et al. suggested that type I mutations were common, and that parental mosaicism was an explanation for some recurrences [12]. The connection between wormian bones and osteogenesis imperfecta has been known for decades. Cremin et al. studied the association between wormian bones and osteogenesis imperfecta, and they introduced the idea of “significance”. They considered wormian bones to be “significant” when there were more than ten of them and especially when they were arranged in a mosaic-like pattern. Based on this study, they suggested that the presence of significant wormian bones could be a strong argument in favor of osteogenesis imperfecta. Based on conventional radiographic interpretations, they concluded that 88% of 81 patients with a diagnosis of osteogenesis imperfecta had a significant number of wormian bones (more than ten). They confirmed that significant wormian bones were not only found in osteogenesis imperfecta but also in other bone dysplasias [13].

Multicentric carpotarsal osteolysis syndrome (MCTO) (OMIM 166300) is a serious bone disorder characterized by bilateral wrist and ankle joint deformities in connection with carpal and tarsal bone osteolysis and progressive nephropathy. The early clinical phenotype can be confused with juvenile idiopathic arthritis. *MAFB mutations* (also known as *V-maf musculoaponeurotic fibrosarcoma oncogene homolog B*) have been identified in all MCTO patients. MAFB, a single-exon gene, is located at 20q12 and encodes the *MAFB* protein, which functions in the activation and differentiation of osteoclasts and development of podocyte foot processes in the renal system. MAFB is a negative regulator of receptor activator of nuclear factor κB ligand—(RANKL—receptor activator of nuclear factor kappa-B ligand) mediated osteoclast differentiation [14]. Reduced *MAFB* expression leads to osteoclast activity, resulting in osteolysis predominantly in the carpal and tarsal bones [15].

## 2. Materials and Methods

The study protocol was approved by the Ethics Committee of the (Ilizarov Scientific Research Institute, No. 4(50)/13.12.2016, Kurgan, Russia). Informed consents were obtained from the patients’ guardians. Seven children and three adults (of 10–28 years) were enrolled in this study. We fully documented these children through detailed clinical and radiological phenotypic characterizations at the clinic of orthogenetics (osteogenetische ambulanz) in the orthopedic Hospital of Speising (Pediatric Department) and through the scientific collaboration of the first author with Ilizarov Center, Kurgan, Russia, collaboration with the Ibn Zohr Institute of Diagnostic radiology in Tunis and scientific collaboration with Policlinic at Národní, Prague, Czech Republic. This study was conducted based on confirmed diagnosis via phenotype and genotype of a group of children and adults and was carried out between 2016 and 2022. We subdivided our patients in accordance with the phenotypic and genotypic diagnostic process.

## 3. Results

### 3.1. Patients with Osteogenesis Imperfecta Type I

Seven unrelated patients (age range from 10–15 years) were diagnosed with osteogenesis imperfecta type I. All demonstrated early natural history of ligamentous hyperlaxity and dentin abnormalities. A lateral skull radiograph of a 10-year-old boy with OI type I found frontal bossing and the abundance of a worm-like phenotype over lambdoid sutures (arrow); note the odontoid hypoplasia (arrow head), atlanto-axial instability and a hypoplastic posterior arch of the atlas. His genotype indicated mutation *COL1A2 p.G1078D*.

Figure 1a, the sagittal reformatted CT scan of the cranium of a 12-year-old boy with OI type I shows massive downward thinning of the occipital area (downward bulging of the occipital lobe of the cerebrum) in connection with progressive softening of the fragile lambdoid sutures (arrow) (Figure 1b). He manifested mutation in *COL1A1*, *showing c.3233 G>A*.

Interestingly, one child of 15 years with no history of fractures was a client of our department because of post-adulthood scoliosis (Cobbs angle of 28°). Surprisingly, he started to complain from bouts of headache, nystagmus, and nocturnal apnea. These symptoms were neglected by other institutes and were treated on symptomatic grounds. Sagittal CT scan of the cranium of this 15-year-old boy with OI type I found bulging of the odontoid process of 6 mm above Chamberlain’s line, signifying basilar impression (arrow). Note the redundancy and the downward bulging of the occipital area (occipital lobe of the cerebrum) in connection with progressive thinning of the lambdoid sutures (arrow head). His genotype demonstrated mutation in *COL1A2*, *pGly634Asp* (Figure 1c).

#### Patients with Osteogenesis Imperfecta Type IVB

Three adult patients of 18, 23, and 28 years all received the diagnosis of OI type IVB. The natural histories of these patients were almost similar. All had short stature, large head, and a history of occasional but not frequent fractures. Interestingly, bowing of the long bones or any other frequent abnormalities did not manifest, as usually seen in patients with OI type IV. All manifested signs of subclinical basilar impression but nevertheless at different ages. The 18-year-old-girl started to experience sudden bouts of intractable headaches associated with nystagmus. She mentioned progressive deterioration in her educational achievement because of headache and poor concentration. The lateral skull radiograph displayed a massive downward redundancy of the occipital area (the occipital part of the skull that contains the occipital cerebrum looks like a bag of worms; arrow) in connection with progressive softening of the lambdoid area. The downward bulging of the lambdoid was obviously overwhelmed with abundant wormian-like bones. Chamberlain’s line displayed 6 mm bulging of the dens into the brain stem. The severity of the upward bulging of the odontoid could be easily assessed via the semi-translation of the atlas onto the occiput (arrow), which resulted in atlanto-axial dislocation. Her genotype demonstrated homozygous mutation in *COL1A2 (p.G322S)* (Figure 2a). 3D reconstruction CT scan superior view of a 23-year-old male patient with OI type IVB demonstrated massive bulging of the occiput, causing tremendous disfigurement of the cranial anatomical structures (Figure 2b).

The 3D reconstruction CT scan of the skull of a 28-year-old patient with OI type IVB demonstrated massive bulging of the occipital area secondary to progressive softening and thinning of the occipital bones, causing a severe bulge over the lambdoid sutures (note the overly stretched pastry-like phenotype), leading to downward shifting of the skull with subsequent upwards ascending of the dens into the brain stem. His genotype exhibited a carrier of the heterozygote mutation in COL1A2, NM_000089.3c.1801>A,p.(Gly601 Ser.) (Figure 3a). The 3D sagittal CT scan of the craniocervical junction displayed immense thinning of the occiput (arrow) and downward bulging around the borders of the lambdoid sutures. Pathological upward bulging of the dens 8 mm above Chamberlain’s line caused the development of subclinical basilar invagination (Figure 3b).

## 4. Multicentric Carpotarsal Osteolysis Syndrome

Episodes of headaches and dizziness were the most bothersome symptomatology in an 8-year-old girl. She was diagnosed with multicentric carpotarsal osteolysis syndrome with a heterozygous mutation of the MAFB gene. Her mutation in *MAFB demonstrated heterozygous mutation* (*c.176>T p.Pro59Leu*).

Neurological examination disclosed dyslexia, alternating strabismus, general hypotonia, and lower limb dysmetria. We started our skeletal survey to further understand the reason behind her unpleasant symptoms. The lateral skull radiograph indicated an 8-year-old girl with multicentric carpotarsal osteolysis syndrome showing multiple wormian bones in the lambdoid sutures and prominence of the occipital bone associated with progressive thinning protuberance of the squamous part. The soft bones of the skull base may have allowed for progressive infolding of the dysplastic clivus and pathological ascending of the dens into the brain stem. Combination with platybasia is a predisposing factor for basilar impression that can lead to severe distortion of the spinal cord and the anterior brain stem (Figure 4a). The 3D reconstruction CT scan of the lambdoid sutures displayed downward redundancy of the lambdoid sutures in a pattern similar to a stretched pastry, which was interpreted via radiology as wormian bones. Note the stretched-pastry-like phenotype (arrows) (Figure 4b).

(Figure 4b). One of our patients was a 10-year-old girl with craniofacial features of (long occipital region, microstomia, and dental malocclusion), arched palate, short and wide distal parts of digits, carpo-tarsal osteolysis, osteoporosis, history of clavicular fracture, and ligamentous hyperlaxity. Her clinical phenotype was consistent with multicentric carpotarsal osteolysis syndrome, her genotype demonstrated heterozygous missense mutations in the *MAFB gene (c.184A>C p.Thr62Pro)*. A 3D reconstruction CT scan was organized to further understand the nature of these wormian bones. The 3D reconstruction CT scan found progressive separation of the lambdoid sutures, giving the appearance of a stretched-pastry-like appearance (arrow head denotes progressive softening and melting of the sutures, in comparison to the normal coronal suture-arrow) (Figure 5a). Sagittal 3DCT of the craniocervical junction found upward bulging of the dens of more than 5.8 mm above Chamberlain’s line *. The overall picture is of upward invasion of the dens into the brain stem (Figure 5b).

We summarized the actual phenotype of the wormian bones via enlarging the 3D reconstruction CT scans of some of the aforementioned patients to illustrate clearly through a 3D reconstruction CT scan of the lambdoid sutures a clear delineation of the typical phenotype of wormian bones. All manifest an overly stretched pastry-like phenotype (arrows) (Figure 6a–c).

## 5. Discussion

In this paper, we focused our efforts on understanding the factual origin of what wormian bones are. We performed extensive conventional and tomographic studies on our current patients. These results are somehow different from what has so far been published in regard to “wormian bones”. The tomographic phenotype of these worm-like lesions in fact is a pathological process of a progressive softening of the lambdoid sutures. The lambdoid sutures are the weight-bearing zone of the skull, and it is well known that the cerebrum is considered the largest and the heaviest part of the brain, consisting of more than 85% of the total brain weight [16,17]. Thereby, the overload of the cerebrum on the weight-bearing zone of the skull (namely the fragile lambdoid sutures) can result in outward and downward bulging of the posterior lower part of the occiput (specifically around the occipital lobe of the cerebrum). Proper assessment of the evolution and the pathological outcome of these wormian bones is imperative (particularly along the lambdoid sutures). The lambdoid sutures are designed to be the weight-bearing joint of the skull in comparison with the other sutures.

Over the last century, several theories appeared in an attempt to detect the etiology behind the development of wormian bones. Riveiro and Von Tschudi postulated that wormian bones occur in connection with a mechanical reason. They suggested that wormian bones are man-made artificial deformities induced in the skulls of children through cranial deformation by adopting mechanical measures [18]. The mechanical measures as described by Riveiro and Von Tschudi are based on the fact that the skulls of infants are soft and liable to mal-manipulation and therefore can be molded in accordance with tribal cultural practices. They believed that wormian bones can occur as a result of this deliberate molding. The phenomena of skull molding in infancy received additional studies [19,20,21]. However, the theory of Riveiro and Von Tschudi has been refuted by Ell-Najjar and Dawson [22].

Others stated that wormian bones may develop in connection with progressive cerebral expansion, and this would explain why they are found in higher numbers in patients with hydrocephalus [23]. Several other studies attempted to discuss the etiology of wormian bones. Some authors explained wormian bones as a manifestation seen in children born with congenital neurological deficits such as hydrocephalus, craniosynostosis, cerebral palsy, and so forth [24,25].

We reviewed the aforementioned papers with great care. Unfortunately, none of the results emerged from genuine recognition of wormian bones. In fact, we encountered a noticeable issue with different authors who, for instance, were confused and mixed up the multi-sutural appearances of the skull, particularly with progressive enlargement of the skull as seen in children with hydrocephalus. We published several papers regarding children with craniosynostosis and all were documented via 3D reconstruction CT scan, though none exhibited wormian bones.

Marti et al. used 3D reconstruction CT scans in order to explain wormian bones. They attributed the reason behind wormian bones as a manifestation of plagiocephaly. They assumed that the cranial asymmetry of the skull leads to pressure exerted on the infant cranial bones, which results in the development of wormian bones. They based their hypothesis on the CT scan images of their group of 12 children with plagiocephaly; the mean number of wormian bones they encountered was (2.33 W bones/CT) [26].

We examined the CT scan images of Marti et al. and we disagree with their conclusion. The images of Marti et al. are of normal sutures rather than genuine wormian bones. The CT scans of Marti et al. are of children with persistent openings of the anterior and posterior fontanels. These occur in children with certain forms of syndromic and non-syndromic disorders. Wormian bones are usually much more abundant and can never exhibit a tiny or a solitary worm-like phenotype.

Multicentric carpotarsal osteolysis is a condition that is characterized by progressive osteolysis of the carpal and tarsal bones and progressive nephropathy, resulting in hypertension and renal failure in early adult life. The tubular bones involved in the foci of osteolysis are said to have a “sucked candy” appearance. Previous reports have described carpotarsal osteolysis associated with peripheral corneal clouding and pulmonary stenosis [27]. Beals and Bird described the association of corneal clouding with osteolysis but without nephropathy [28]. Malecha et al. similarly focused on carpotarsal osteolysis associated with bilateral corneal clouding in patients described as being diagnosed with idiopathic multicentric osteolysis [29].

Basilar impression/invagination is a serious deformity of the craniocervical junction. It may lead to a life-threatening situation because of stenosis of the foramen magnum and compression of the medulla oblongata. Sudden death or severe neurological deficits due to brain stem compression, particularly in patients with sudden unstable head–neck movements or jarring can occur [30].

This is a type of vertical atlanto-axial instability, or type 2 according to Goel’s classification. There is no dislocation of the C2 dentoideum into the vertebral canal [31].

## 6. Conclusions

The weight of the cerebrum exerts a direct load on the loose and soft boney junction/joints of the occiput as presented by the lambdoid sutures. The reconstruction CT scans of our current patients exhibited typical softening and melting of the lambdoid sutures. This overall process is somehow similar to an overly stretched soft pastry. The risk of progressive softening of the lambdoid is not restricted to the distorted anatomy of the cranium, but also includes the total derangement of the craniocervical junction. Basilar impression/invagination is a serious complication of the craniocervical junction. It may lead to a life-threatening situation because of stenosis of the foramen magnum and compression of the medulla oblongata. Sudden death or severe neurological deficits due to brain stem compression can occur. Pediatric spine surgeons should ensure regular follow-ups if neurological deficits worsen. In this case, decompression operation might be an option. The term wormian is not just a term given to skulls; in fact, it is a more complicated pathological process along the lambdoid sutures occurring because of the progressive softening of the lambdoid sutures, which are considered the weight-bearing zone of the skull /brain and specifically the occipital lobe of the cerebrum.

## Figures and Tables

**Figure 1 diagnostics-13-00874-f001:**
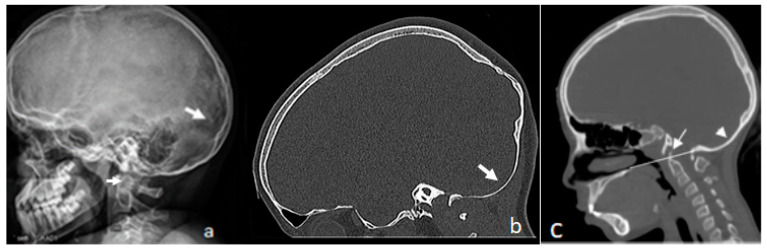
(**a**–**c**) Lateral skull radiograph of the 10-year-old boy with OI type I showing frontal bossing and abundance of worm-like phenotype over lambdoid sutures (arrow); note the odontoid hypoplasia (arrow head), atlanto-axial instability and hypoplastic posterior arch of the atlas. His genotype *exhibited mutation COL1A2 p.G1078D* (**a**). Sagittal reformatted CT scan of the cranium of a 12-year-old boy with OI type I showing massive downward thinning of the occipital area (downward bulging of the occipital lobe of the cerebrum) in connection with progressive softening of the fragile lambdoid sutures (arrow). He manifested *mutation in COL1A1*, *showing c.3233 G>A* (**b**). Sagittal CT scan of the cranium of a 15-year-old boy with OI type I showing bulging of the odontoid process of 6 mm above Chamberlain’s line, signifying basilar impression (arrow). Note the redundancy and downward bulging of the occipital area (occipital lobe of the cerebrum) in connection with progressive thinning/softening of the lambdoid sutures, arrow head). His genotype demonstrated mutation in *COL1A2,pGly634Asp* (**c**).

**Figure 2 diagnostics-13-00874-f002:**
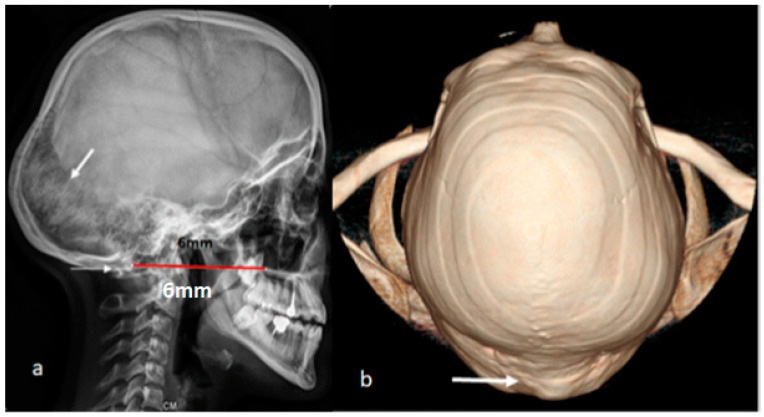
(**a**,**b**) Lateral skull radiograph of an 18-year-old girl with OI type IV B showing a massive downward redundancy of the occipital area along the lambdoid sutures (the occipital part of the skull that contains the occipital cerebrum looks like a bag of worms, arrow) in connection with progressive softening of the lambdoid area. The latter was obviously overwhelmed with an abundancy of wormian-like bones. Chamberlain’s line indicates 6 mm bulging of the odontoid into the brain stem. The severity of the upward bulging of the dens can be easily assessed via the semi-translation of the atlas onto the occiput (arrow), which resulted in atlanto-axial dislocation. Her genotype demonstrated *homozygous mutation in COL1A2 (p.G322S)* (**a**). Three-dimensional reconstruction CT scan superior view of a 23-year-old male patient with OI type IVB showing massive bulging of the occiput, causing tremendous disfigurement of the cranial anatomical structures. His genotype demonstrated mutation in *COL1A2 c.2827G>A* (**b**).

**Figure 3 diagnostics-13-00874-f003:**
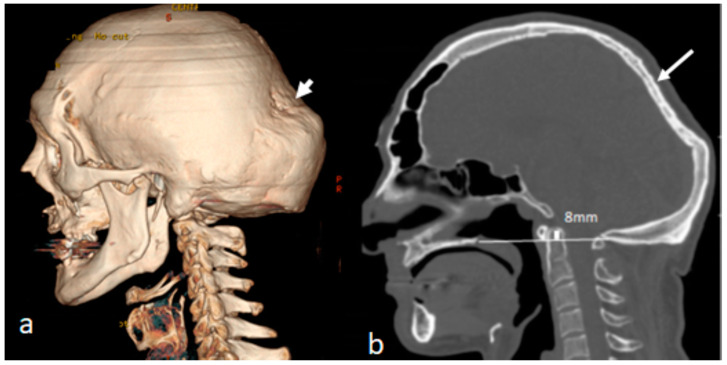
(**a**,**b**) Three-dimensional reconstruction CT scan of the skull of a 28-year-old patient with OI type IVB showing the massive bulging of the occipital area secondary to progressive softening and thinning of the occipital bones, causing a severe bulge over the lambdoid sutures (note the overly stretched pastry-like phenotype—arrow) leading to downward shifting of the skull with subsequent upwards ascending of the dens into the brain stem. His genotype exhibited a carrier of the heterozygote mutation of *COL1A2*, *NM_000089.3c.1801>A,p.(Gly601 Ser.)* (**a**). Three-dimensional sagittal CT scan of the craniocervical junction showing immense thinning of the occiput (arrow) and downward bulging around the borders of the lambdoid sutures. Pathological upward bulging of the dens 8 mm above Chamberlain’s line causing the development of subclinical basilar invagination (**b**).

**Figure 4 diagnostics-13-00874-f004:**
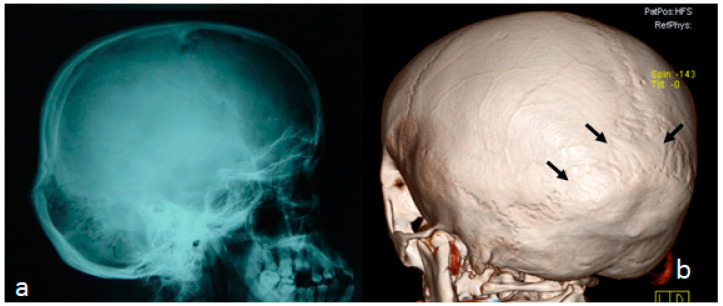
(**a**,**b**) Lateral skull radiograph showing an 8-year-old girl with multicentric carpotarsal osteolysis syndrome showing multiple wormian bones in the lambdoid sutures and prominence of the occipital bone associated with progressive thinning protuberance of the squamous part. The soft bones of the skull base might allow for progressive infolding of the dysplastic clivus and translocation of the odontoid into the posterior fossa. Combination with platybasia is a predisposing factor for basilar impression and can lead to severe distortion of the spinal cord and the anterior brain stem (**a**). Three-dimensional reconstruction CT scan of the lambdoid sutures showing downward redundancy of the lambdoid sutures in a pattern similar to a stretched pastry, which was interpreted via radiology as wormian bones. Note the stretched-pastry-like phenotype (arrows) (**b**).

**Figure 5 diagnostics-13-00874-f005:**
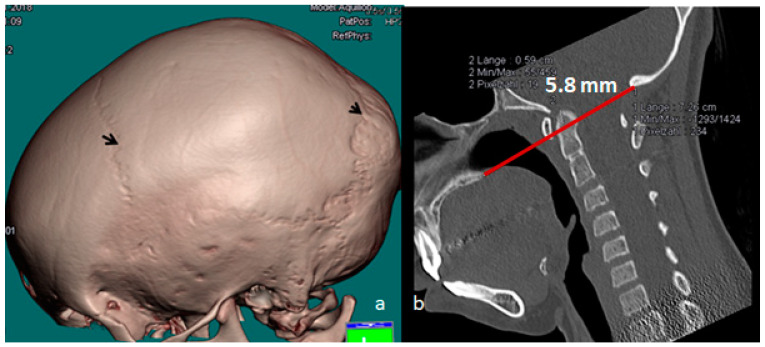
(**a**,**b**) Three-dimensional reconstruction CT scan of a 10-year-old girl with multicentric was organized to further understand the nature of these wormian bones. Three-dimensional reconstruction CT scan shows progressive separation of the lambdoid sutures, giving an overly stretched-pastry-like appearance (arrow head denotes progressive softening and melting of the sutures, in comparison to the normal coronal suture-arrow) (**a**) Sagittal 3DCT of the craniocervical junction shows Chamberlain line (which joins the hard palate to the posterior lip of the foramen magnum). The odontoid process length is greater than 5.8 mm. The overall picture is of dolicho-odontoid with posterior inclination (**b**) * Chamberlain line: a line connecting the back of the hard palate with the opisthion. Seen on a lateral view of the craniocervical junction via conventional radiograph, but with much better visualization and localization via 3D sagittal CT scan of the craniocervical junction has been applied. The importance lies in diagnosing basilar impression or invagination, particularly when the tip of the dens is more than 3 mm above the line.

**Figure 6 diagnostics-13-00874-f006:**
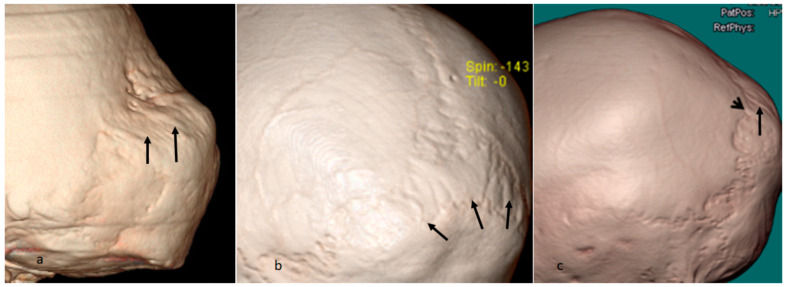
(**a**–**c**) We summarized the actual phenotype of the wormian bones via enlarging the 3D reconstruction CT scans of some of the aforementioned patients to illustrate clearly through 3D reconstruction CT scan of the lambdoid sutures a clearly delineation of the typical phenotype of wormian bones. All manifest an overly stretched pastry-like phenotype (arrows).

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
