# Peer review of "The Tomographic Study and the Phenotype of Wormian Bones"

_diagnostics, 2023, doi:10.3390/diagnostics13050874_

Round 1
Reviewer 1 Report
Please see note below
This manuscript is important to the field of Wormian Bones, however it needs some revision.
Line 22: use "which" instead of who; Line 23: "and so forth" is meaningless in a manuscript, should be deleted. Line 25: and "are" attempting. Line 27: remove second "diagnosis of". Line 32: "concerning" not concerned. Line 33: "responsible for" instead of connected. Line 35: remove the second "patients". Line 41: "This results", instead of which results. Line 45: The latter causing "a" pathological. Line 56: there are "at" least. Line 62 "represent" instead of represents. Line 86-87, wrong spelling of author (see references), and I do not see Glorieux listed in references.
Make sure that the full meaning of abbreviations , e.g. MCTO, others, have been written in full. Line 165: Remove "nevertheless. Line 170: Remove extra parenthesis. Line 242: remove the first "showed". Line 273: "result" instead of results. Line 278: in comparison "to" instead of with sounds better. Line 281: "connection with a mechanical reason "explain". Line 291: Line 291: "reviewed" instead of revised. Line 292: none of the results of various studies have emerged from genuine recognition of wormian bones. Line 341: a term given to the skulls, "but "in fact ------.
This manuscript has a great deal of merit, however it needs some changes. For example, "there are two sections on Materials and Methods which is not necessary and can be confusing. Similarly with Conclusions. Perhaps three lines in the "Abstract Background" may be sufficient as preliminary, then expanded in their relevant sections.
The images are excellent and very clear. They are very important highlights of this manuscript.
I would also like to see a little more references, mainly some recent references which are available in the literature.
Author Response
Dear Reviewer,
Thank you so much for your constructive remarks.
our response is attached.
Warmest regards
Al Kaissi

Reviewer 2 Report
The submission is a complete manuscript entitled "The Tomographic Study and the Phenotype of Wormian Bones".
This is a well-written, tidy and well-structured manuscript. The authors clearly explain the purpose of the study.
After carefully evaluating the manuscript, I have only some minor recommendations that I hope the authors take into account before publication. I hope the authors take my comments as constructive advice.
Keywords. Two keywords are already in the article title: "Phenotype"; "Wormian Bones". Please, consider using other terms.
Line 20-21. The authors mention, "The principal complaint for the pediatric group were". Please, provide the same information for the adult group.
Line 27. Is there a double space between "of osteogenesis"? If that is the case, please correct this typo. The same happens in line 35 and along the text.
Line 40. Please, for consistency, add a hyphen when referring to "worm like".
Line 40-42. Please, rephrase to improve readability.
Introduction
Please, for consistency, decide if using "Wormian" (with uppercase) or "wormian" (with lowercase).
If the authors intend to use the acronym OI to refer to "Osteogenesis imperfecta",. Please, use both terms together the first time, e.g., "Osteogenesis imperfecta (OI)", and only use OI afterwards.
Line 85-86 Please, rephrase this sentence. Are the authors asking a question or making an affirmation?
Please provide further explications about OI Type II and III.
Materials and Methods
Although in the results section, further details about the patient's demographic data are provided. It would be helpful to find this information in this section in the main text or a table.
The authors should provide all relevant information about the CT data acquisition (e.g., resolution, matrix, among others) and the software used for the 3D reconstruction CT.
Results
Please consider moving reference to "Figure 1a" to line 130 to improve interpretation if that is the case.
Figure 1a. Using the "(a); (b); (c)" at the end of the sentences affects readability. Please, consider moving this to the end of each sentence: e.g. (a)Lateral skull radiograph.
The white arrow in Figure 1a is challenging to identify. Please, consider using another colour.
Please, provide details about type IVB in the Introduction.
Figure 2. Please use another colour to visualize the label indicating the 6 mm bulging of the odontoid.
Line 94. Did the authors mean "odontoid" instead of "odntoid"?
Calling figures at the beginning of a sentence, using parentheses and lowercase (e.g., 125 and 229), affects readability. Please, change this.
Figure 5a. Looking at the figure, one cannot note the "progressive separation of the lambdoid sutures giving the appearance of a stretched pastry-like", as indicated by the authors. Please, consider improving the image quality, enhancing contrast, or using another view.
The text in Figure 5b is not readable. The authors should consider improving the image quality.
The authors mention multiple times in the text "Chamberlain's line". However, a description is not provided until line 260. Please, consider moving this explanation to the Introduction.
Discussions
The authors should consider critically evaluating key areas addressed and derive conclusions from the results. A more elaborated integration of the results obtained with published literature would be necessary.
Author Response
Dear Reviewer,
We mostly responded to the most of your comments. A nother set of 3D reconstruction showed the typical overly streched pastry like phenotype of the wormian bones has been added as figure 6 a,b,c.
Warmest regards
Al Kaissi
